# Observing electron localization in a dissociating $H_2^+$ molecule in real time

H. Xu[1], Zhichao Li[2], Feng He[2], X. Wang[1,3], A. Atia-Tul-Noor[1], D. Kielpinski[1], R.T. Sang[1] & I.V. Litvinyuk[1]

Dissociation of diatomic molecules with odd number of electrons always causes the unpaired electron to localize on one of the two resulting atomic fragments. In the simplest diatomic molecule $H_2^+$ dissociation yields a hydrogen atom and a proton with the sole electron ending up on one of the two nuclei. That is equivalent to breaking of a chemical bond—the most fundamental chemical process. Here we observe such electron localization in real time by performing a pump–probe experiment. We demonstrate that in $H_2^+$ electron localization is complete in just 15 fs when the molecule's internuclear distance reaches 8 atomic units. The measurement is supported by a theoretical simulation based on numerical solution of the time-dependent Schrödinger equation. This observation advances our understanding of detailed dynamics of molecular dissociation.

[1] Centre for Quantum Dynamics and Australian Attosecond Science Facility, Griffith University, Nathan, Queensland 4111, Australia. [2] Key Laboratory for Laser Plasmas (Ministry of Education) and School of Physics and Astronomy, Collaborative Innovation Center of IFSA (CICIFSA), Shanghai Jiao Tong University, Shanghai 200240, China. [3] School of Nuclear Science & Technology, Lanzhou University, Lanzhou 730000, China. Correspondence and requests for materials should be addressed to H.X. (email: hanxu1981@gmail.com) or to F.H. (email: fhe@sjtu.edu.cn) or to I.V.L. (email: i.litvinyuk@griffith.edu.au).

One of the ultimate goals of strong-field laser physics is to probe and control the dynamics of laser-induced chemical reactions. In the past decade, it has been shown both experimentally[1–7] and theoretically[8–11] that the carrier-envelope phase (CEP)[12] of a few-cycle pulse can be an important knob for controlling laser-induced chemical reactions, for example, dissociation of a hydrogen molecular ion $\left(H_2^+ \rightarrow H^+ + H\right)$, where the electron can preferentially end up with either one of the two protons. The dynamics of electron localization during the $H_2^+$ dissociation can be intuitively understood by following the evolution of the double-well molecular potential as the internuclear distance $(R)$ increases. For short $R$ (near the equilibrium separation) the intramolecular potential barrier is low and the shared electron can freely move between the nuclei driven by the oscillating laser electric field. With increasing $R$, the potential barrier between the two protons becomes higher and wider, and the electron motion becomes more difficult, until the electron eventually gets stuck on one of the protons. By changing the CEP of the few-cycle pulse one can steer the direction of electron localization, with experimental asymmetries of over 50% reported with this type of control[1,6]. Apart from controlling the outcome of a laser-driven chemical reaction, observing its dynamics in real time is also of great fundamental interest as it could help us understand the most fundamental physical mechanisms behind the laser–molecule interactions. For example, a molecular movie of proton migration during the isomerization process in the acetylene cation has been demonstrated recently[13]. Molecular dissociation, resulting in breaking of a chemical bond, is another process whose detailed dynamics is of fundamental interest.

In this article, we answer the question: how fast and at what internuclear distance does the electron localize and the chemical bond break in dissociating $H_2^+$? To answer this question we perform a pump–probe experiment that aims not only to control the final product, but also to time-resolve the dynamics of electron localization during the $H_2^+$ dissociation by measuring the delay-dependent asymmetry of proton emission. Both pump and probe pulses are only 5 fs in duration and have a controlled CEP. The more intense pump pulse ionizes a neutral $H_2$ molecule and initiates dissociation of $H_2^+$. The much weaker time-delayed probe pulse drives the electron motion in the dissociating molecule. By measuring the asymmetry of proton emission as a function of pump–probe delay we demonstrate that direction of electron localization can be controlled by the probe pulse only within the first 15 fs following ionization of $H_2$. For larger pump–probe delays direction of electron localization is completely determined by CEP of the pump pulse. With support from a numerical simulation, we conclude that in $H_2^+$ electron localization is complete in just 15 fs when the molecule's internuclear distance reaches 8 atomic units.

## Results

**Experimental scheme.** Our experimental setup is shown in Fig. 1a. The CEP-locked linearly polarized $\sim 5$ fs few-cycle pulse with central wavelength of $\sim 750$ nm is diverted into a Mach–Zehnder interferometer to produce pump–probe pulse pair with a controllable delay. The pulses are tightly focused inside the Reaction Microscope (REMI) apparatus on a neutral $H_2$ supersonic jet, and the full three-dimensional momenta of the ion fragments are measured by the REMI. The polarization axes of the pump and probe pulses are both parallel to the time-of-flight axis (defined as $z$ axis) and normal to propagation direction of the laser beam ($x$ axis) and that of gas jet ($y$ axis). The dissociative ionization process of neutral $H_2$ in the few-cycle pump–probe field is illustrated in Fig. 1b. At the peak of the

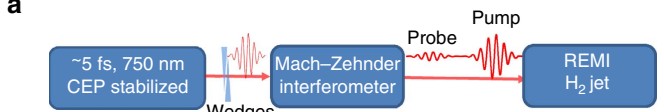

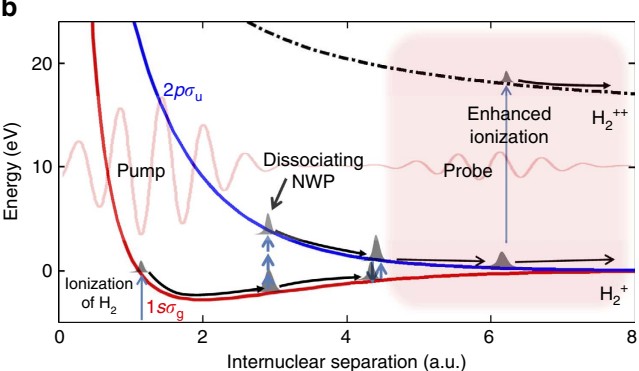

**Figure 1 | Experimental scheme.** (**a**) Experimental scheme. The laser pulse has duration of 5 fs with central wavelength of 750 nm, and the estimated carrier-envelope phase (CEP) noise (root mean square) is <360 mrad. A pair of fused silica wedges is used to control the CEP of the few-cycle pulse. A Mach–Zehnder interferometer is used to create pump–probe pulse pair with controllable delay. The pulse pair is then sent into reaction microscope (REMI) to interact with a supersonic $H_2$ jet, where the momentum of the resulting charged particles are measured. (**b**) An intuitive physics picture for describing the dissociative ionization process of $H_2$ in the pump–probe pulse pair laser field. The neutral hydrogen molecule is ionized at the peak of the pump pulse and then excited into a dissociative state either radiatively or by electron recollision. The probe pulse is sent in to interact with the dissociating $H_2^+$, where $R$ of $H_2^+$ can be controlled by tuning the delay between pump and probe pulses. The probe pulse can steer the electron motion during the dissociation process that leads to a significant delay-dependent asymmetry of the electron localization. On the other hand, the probe pulse can also induce a further enhanced double ionization of the dissociating $H_2^+$ at critical internuclear separations, and the kinetic energy release of the enhanced ionization channel is measured for analysing the propagation dynamics of the dissociating nuclear wavepackets (NWPs).

pump pulse, neutral $H_2$ is ionized and a nuclear wavepacket (NWP) is launched onto $1s\sigma_g$ potential curve of $H_2^+$. After the initial ionization $H_2^+$ can be further dissociated by the tail of the pump pulse radiatively coupling $1s\sigma_g$ and $2p\sigma_u$. The $R$ of the dissociating $H_2^+$ will gradually increase with the bound electron migrating between two nuclei until $R$ is too large and the electron gets trapped on one of the protons. The delayed CEP-stabilized probe pulse, introduced during the dissociation process, is used to drive the electron motion and affect the direction of electron localization. As we have shown in our previous work with random CEP pulses, the probe pulse can also cause enhanced ionization (EI)[14–16] of $H_2^+$ when the internuclear separation $R$ of dissociating NWP reaches critical distance, and the delay-dependent kinetic energy release (KER, the sum of kinetic energy of two protons from the same $H_2^+$) of EI channel is used for tracing time evolution of $R$ in dissociating $H_2^+$.

**Asymmetry of proton emission.** The asymmetry parameter for proton emission is used to quantitatively characterize the degree and direction of electron localization. This parameter can be expressed as,

$$A = \left(N_{\text{up}} - N_{\text{down}}\right) / \left(N_{\text{up}} + N_{\text{down}}\right),$$

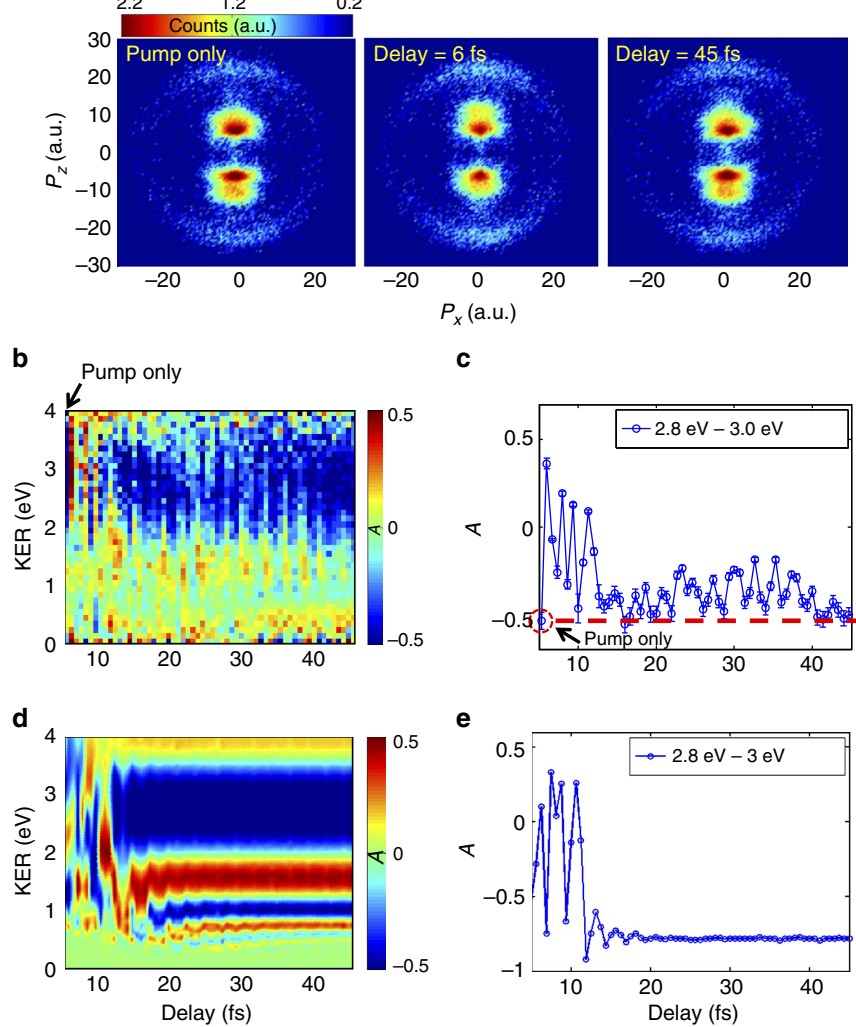

**Figure 2 | Proton momentum and asymmetry.** (**a**) Measured proton momentum distribution in laser polarization plane ($P_x$–$P_z$ plane) with pump pulse only (left), for pump–probe delay of 6 fs (middle) and for delay of 45 fs (right). Measured (**b**) and calculated (**d**) delay-dependent energy-resolved asymmetry of proton emission. The data for zero delay (indicated by the arrow in **b**) are measured with the pump pulse only. Measured (**c**) and calculated (**e**) delay-dependent asymmetry of proton emission for kinetic energy release (KER) from 2.8 to 3.0 eV. The red dashed line in **c** indicates the value of the asymmetry measured with pump pulse only. Error bars for measured asymmetry are obtained by propagating the estimated shot noise of the corresponding counts.

where $N_{up}$ ($N_{down}$) is the number of protons ejected into up (down) direction along the laser polarization axis. It has been shown that a few-cycle pulse with the optimal intensity of $6 \times 10^{14}\,\mathrm{W\,cm^{-2}}$ can produce a strong CEP-dependent asymmetry modulation ($>0.5$) for protons in kinetic energy region (2–3 eV) where the above threshold dissociation and the three-photon dissociation channels overlap[6]. In this experiment, the pump pulse is set to the same optimal intensity of $\sim 6 \times 10^{14}\,\mathrm{W\,cm^{-2}}$ to maximize the CEP-dependent asymmetry modulation, and we observe an asymmetry modulation with amplitude of $\sim 0.5$ induced by the pump field alone. During the delay scan the CEP is locked to the value at which the pump pulse by itself produces negative maximum value of the asymmetry parameter. A weak probe intensity of $\sim 6 \times 10^{13}\,\mathrm{W\,cm^{-2}}$ is chosen that is high enough to change the direction of electron localization, and meanwhile low enough to avoid depletion of dissociating $H_2^+$ by EI.

The measured energy-resolved asymmetry as a function of delay is shown in Fig. 2b, and the data that are taken with the pump pulse only are shown on the far left side of the figure for comparison. The spectrum is characterized by a strong

modulation of the asymmetry and change of its direction for delay times of $<15$ fs, and a much weaker modulation and constant negative direction of the asymmetry for larger delays. To demonstrate this delay-dependent asymmetry modulation more clearly, we focus on a narrow KER region of $2.8\,\mathrm{eV} < \mathrm{KER} < 3\,\mathrm{eV}$ (see Fig. 2c). The first data point ($\tau = 0$) shows asymmetry of $-0.5$ with pump pulse only, meaning that the downward proton emission is dominating. The asymmetry oscillates rapidly as a function of delay with the periodicity of the laser field, agreeing well with the observation in extreme-ultraviolet infrared pump–probe experiment[4]. For the delay of 6 fs, the asymmetry changes from its original value of $-0.5$ to 0.4, indicating that the direction of electron localization is completely reversed by the probe pulse. This delay is almost equal to the pulse duration of 5 fs, so the probe pulse overlaps with the tail of the pump pulse. Therefore, it is not surprising that the temporal profile of the electric field is sufficiently modified to completely reverse the direction of electron localization. The delay-dependent asymmetry modulation remains strong until the delay reaches 15 fs, at which point it suddenly dramatically weakens and asymmetry no longer changes its direction. At the end of the scan

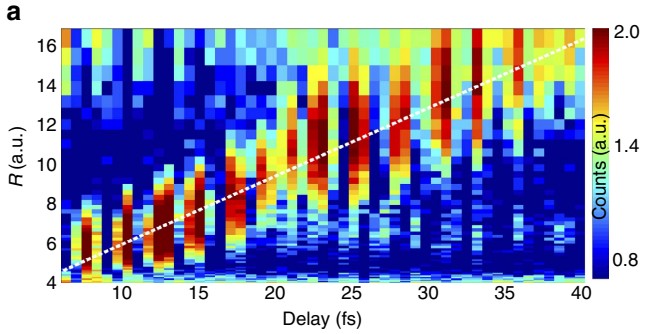

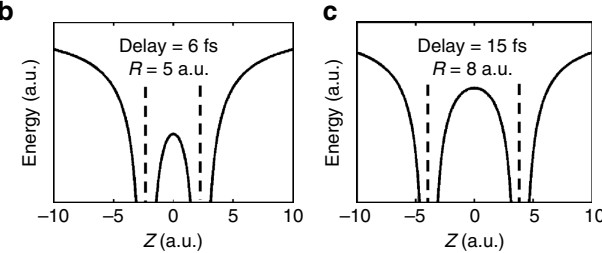

**Figure 3 | Internuclear separation of electron localization.** (**a**) Retrieved time evolution of the dissociating nuclear wavepackets (NWPs) from the measured delay-dependent kinetic energy release (KER) of enhanced ionization channel by using $KER(\tau) = \frac{1}{R(\tau)} + 0.7\,eV$. Since the measured KER of the bond softening channel is 0.7 eV, we set the initial kinetic energy of dissociating NWP to 0.7 eV. The overlayed white dashed line is a linear fit for the peak of the propagating dissociating NWP. (**b,c**) The instantaneous Coulomb potential curve of the dissociating $H_2^+$ for $R$ of 5 a.u. when delay is 6 fs (**b**), and $R$ of 8 a.u. when delay is 15 fs (**c**) respectively. The potential curve is calculated by using a traditional soft-core potential formula of $V(z,R) = \frac{1}{\sqrt{(z+R/2)^2 + a^2}} + \frac{1}{\sqrt{(z-R/2)^2 + a^2}}$, where $a$ is the softening parameter.

($\tau = 45\,fs$), the downward proton emission dominates again (Fig. 2a, right column) and the asymmetry goes back to the original value of $-0.5$.

This behaviour clearly reflects the real-time dynamics of electron localization in dissociating $H_2^+$. At short delays ($\tau < 15$ fs) the internuclear distance remains small and the shared electron can migrate between the two nuclei when driven by a relatively weak electric field, so that the probe pulse could modify the outcome of dissociation. At longer delays ($\tau > 15$ fs) the internuclear distance becomes too large and the inner potential barrier too wide and high for the electron to complete the migration within the half-cycle of electric field oscillation and it effectively gets trapped on one of the nuclei—the localization is complete with its direction determined by the CEP of the pump pulse alone.

**Numerical model.** To validate and interpret the experimental observations, we performed the time-dependent Schrödinger equation simulation (see Methods for details) with the experimental laser parameters. In our numerical model, both the electronic and nuclear motions are confined along the laser polarization direction, and the ionized electron flux is absorbed by the boundary by adopting a mask function on the boundary of the simulation box. Figure 2d shows the calculated energy-resolved asymmetry as a function of delay. The asymmetry in the KER region 2.8 eV < KER < 3 eV is shown in Fig. 2e. The simulation and the measurement demonstrate qualitatively the same behaviour—for delays under 15 fs the asymmetry is strongly affected by the probe pulse, while for longer delays its direction remains fixed and determined by the

pump pulse alone. We performed simulations for a number of different pump (from 2 to $6 \times 10^{14}\,W\,cm^{-2}$) and probe (from $3 \times 10^{13}$ to $2 \times 10^{14}\,W\,cm^{-2}$) pulse intensities. With probe intensities varying over almost an order of magnitude range, all simulations showed the same qualitative behaviour and the same 15 fs cut-off time for electron localization. The robustness of these results indicates that we observe a signature of a fundamental dynamical process rather than an intensity-specific effect. We conclude that after 15 fs electron localization in dissociating $H_2^+$ is complete and cannot be reversed by applying another laser pulse.

**Asymmetry at larger delays.** One may note that the experimental asymmetry parameter still exhibits noticeable delay-dependent oscillations until $\tau = 40$ fs as shown in Fig. 2c. At those larger delays, the double ionization due to EI becomes a significant channel even at low probe intensities. EI produces a pair of protons with kinetic energy similar to that of the dissociative channel. Since our analysis cannot completely exclude those double ionization protons (see Methods for details), which show no asymmetry but are also counted in the asymmetry calculation, the resulting overall asymmetry will be less than that for the pure singly ionized channel. In addition to that, the delay-dependent EI yield modulation could lead to the observed delay-dependent modulation of asymmetry. It is also possible that EI itself exhibits a spatial asymmetry leading to a selective depletion of $H_2^+$ by a spatially asymmetric CEP-locked probe pulse in a manner described in ref. 17 for $I_2$ with spatially asymmetric two-colour pulses. Such selective depletion could affect the experimental asymmetry parameter even in a cleanly separated dissociation channel. Neither of these effects is included in our model asymmetry calculations.

**Internuclear distance of electron localization.** It is also interesting to measure the internuclear distance at which electron localization takes place. This is achieved by retrieving the dynamics of dissociating NWPs from the measured delay-dependent KER spectrum of the double-ionization channel. When the $R$ of dissociating $H_2^+$ approaches the so-called critical distance for EI, the tunnelling ionization rate can increase dramatically[14]. The weak probe can then promote part of the dissociating nuclear wave packet to the repulsive potential curve of $H_2^{2+}$ and then a pair of energetic protons will be produced via the Coulomb explosion process. The KER of that channel is given by the instantaneous position $R$ of the dissociating NWP when the probe pulse arrives that can be formally expressed as $KER(\tau) = 1/R(\tau) + E_0$, where $\tau$ is delay and $E_0$ is the initial kinetic energy obtained in dissociation process. Due to the very short pulse duration and extreme nonlinearity of tunnelling ionization the nuclear motion is nearly frozen during the probe-induced EI, making it possible to resolve the position of NWP for different delay, that is, $R(\tau)$. Figure 3a shows the measured time evolution of dissociating NWP by converting the measured delay-dependent KER spectrum to delay-dependent $R$ distribution. Assuming the dissociating NWP is propagating at a constant velocity, we perform a linear fit for the time evolution of the peak of dissociating NWP, as shown by the white dashed line in Fig. 3a. Figure 3b shows the instantaneous Coulomb potential of the dissociating $H_2^+$ for time delay of 6 fs, where the $R$ (fit value of peak position of NWP) reaches $\sim 5$ a.u. For such a small internuclear separation the inner barrier is low, so that the electron can still follow the oscillation of the driving laser field and lead to the observed strong delay-dependent asymmetry modulation. Internuclear distance $R$ reaches $\sim 8$ a.u. at delay of 15 fs. The corresponding Coulomb potential (Fig. 3c) shows a

much higher and wider inner barrier that the electron can no longer transverse. The electron localization completes and the chemical bond is broken when protons in $H_2^+$ are separated by the distance of 8 a.u.

## Discussion

In summary, pump–probe experiment with CEP-locked few-cycle pulses is used to control the electron localization in dissociating $H_2^+$ and to observe electron localization and breaking of a chemical bond in real time. The very short pulse duration allows us to resolve the response of the bound electron to external driving field at different internuclear separations during the dissociation process. Similar to previous extreme-ultraviolet infrared pump–probe experiment[4] that demonstrate the electron localization control for high-KER dissociation channels (10 eV < KER < 20 eV), we show that the infrared–infrared pump–probe method can control the asymmetry of the electron localization for low KER channels (KER < 4 eV) by tunning the delay between pump and probe pulses. The asymmetry modulation is strong and its direction can be reversed for delays under 15 fs. By retrieving time-dependent distribution of internuclear separations $R(t)$ in dissociating $H_2^+$ we determined that in that molecule in our experimental condition electron localization happens 15 fs after the initial ionization of neutral $H_2$ when the internuclear separation reaches 8 a.u. That is when and where the chemical bond is broken and the molecule fragments into a proton and a hydrogen atom.

## Methods

**Experimental details.** The experimental apparatus includes a 1 kHz CEP-stable few-cycle laser system (Femtopower Compact Pro, Femtolasers) with hollow-core fibre compressor, and a REMI apparatus[18,19]. The CEP of the few-cycle pulse from fibre compressor is measured by an f-2f interferometer, and used as a feedback signal for CEP stabilization feedback loop. The estimated CEP noise is < 360 mrad (root mean square) when the feedback loop is closed, and the CEP locking can be stable for ~ 2 h, typically. A pair of fused silica wedges is used to control the CEP and to compensate chirp of the few-cycle pulses. A Mach–Zehnder interferometer that is designed for few-cycle laser pulses is used to produce the pump–probe pulse pair. A motorized stage can generate time delay with subcycle precision. The intensities of pump and probe pulses are independently controlled by irises, and the *in situ* peak intensities of pump and probe pulses are measured by using the ion-recoil momentum method[20,21]. The pulse pair is tightly focused onto a supersonic $H_2$ gas jet by a zero group delay dispersion concave mirror ($f = 75$ mm), and the generated protons are detected by a time- and position-sensitive delay-line anode detector in REMI that allows precise measurement of three-dimensional momentum vector for each proton. The zero delay between pump and probe is determined by measuring the delay-dependent ionization of Ne gas jet, where $Ne^+$ yield can maximize at zero delay that corresponds to the highest peak intensity. To preferentially select those protons from the dissociation channel $(H_2^+ \rightarrow H + H^+)$, we use a filter that discards ionization events with more than one ion per laser shot detected. On the other hand, to highlight the enhanced ionization channel, we apply a momentum conservation filter that chooses those ionization events with two ions per laser shot and the sum of their momentum is < 5 a.u., that is, $|P_{ion1} + P_{ion2}| < 5$ a.u., ensuring that the selected proton pair originates from a single hydrogen molecule. It should be noted that we cannot completely distinguish between protons of the same energy produced by single and double ionization channels.

**Numerical simulation.** We numerically solved the time-dependent Schrödinger equation (atomic units are used throughout unless indicated otherwise)

$$i\frac{\partial}{\partial t}\psi(z,R;t) = \left[\frac{p_R^2}{2\mu} + \frac{1}{2}(p_z + A(t))^2 + V(z,R) + \frac{1}{R}\right]\psi(z,R;t),$$

where $\mu$ is the reduced nuclear mass, $p_R$ and $p_z$ are nuclear and electronic momentum operators, $A(t)$ is the laser vector potential and $A(t) = -\int_0^t dt' E(t')$ with $E(t')$ being the laser electric field. The Coulomb potential is written as

$$V(z,R) = -\sum_{s=\pm 1} \frac{1}{1/\alpha(R) - \alpha(R)/5 + \sqrt{(z+sR/2)^2 + (\alpha(R)/5)^2}},$$

where $\alpha(R)$ is the $R$-dependent soft core function that is determined by fitting the calculated energies of the ground and the first excited state of $H_2^+$ to their real Coulomb values. That function can be closely approximated by $\alpha(R) = 0.9$

$+ 0.5806 \times \exp(-0.5876 R)$ and it monotonously decreases from 1.5 to 0.9 as $R$ increases. The simulation box extends from $-1,000$ to $1,000$ a.u. along $z$ dimension and from 0 to 100 a.u. along $R$ dimension, and the spatial and time steps are $\Delta z = 0.3$ a.u., $\Delta R = 0.02$ a.u. and $\Delta t = 0.2$ a.u. A mask function $\cos^{1/6}$ is set on the boundary for suppressing the unphysical reflection. At the terminal time $t_f$, we Fourier transformed the wave function in the range $R > 10$ a.u. and $|z \pm \frac{R}{2}| < 20$ a.u., and obtained the upgoing and downgoing dissociative wavepackets in the momentum representation. After that, the energy-resolved asymmetry can be extracted by following the same procedure applied in the experiment.

In simulations, the initial wave function is the superposition of the first 10 lowest molecular vibrational states of $H_2^+$, weighted by the Franck–Condon factors. Then, we applied a $\sin^2$ laser pulse to dissociate $H_2^+$ that plays a role of the pump pulse in the experiment. The probe pulse is introduced at different time delays to steer the electron localization. Similar to the experimental conditions, both the pump pulse and the probe pulse sit on the pedestal with long duration but a much lower intensity. In the simulations, we used the combined laser fields as following:

$$E = E_{pedestal}\cos(\omega t) + E_{pump}\cos(\omega t)f_{pump} + E_{probe}\sin[\omega(t-\Delta t)]f_{probe}.$$

To exactly simulate the experimental conditions the pump pulse had the field amplitude $E_{pump} = 0.13$ ($6 \times 10^{14}$ W cm$^{-2}$) and the envelope $f_{pump} = \cos^2[\pi t/5T]$, $-2.5T < t < 2.5T$. The probe pulse had the following parameters $E_{probe} = 0.041$ ($6 \times 10^{13}$ W cm$^{-2}$), $f_{probe} = \cos^2[\pi(t-\Delta t)/5T]$, $-2.5T < t - \Delta t < 2.5T$. The pedestal had the constant amplitude $E_{pedestal} = 0.013$ ($6 \times 10^{12}$ W cm$^{-2}$). We also performed simulations for a number of different pump and probe intensities, with pump intensities varying from $2 \times 10^{14}$ to $6 \times 10^{14}$ W cm$^{-2}$ and probe intensities varying from $3 \times 10^{13}$ to $2 \times 10^{14}$ W cm$^{-2}$. The qualitative behaviour was very similar for all intensities tested. Note that the time delay in Fig. 2d,e corresponds to $\Delta t + 2.5T$, that is, the delay between the birth of $H_2^+$ and the probe pulse.

**Data availability.** The data that support the findings of this study are available from the corresponding authors on request.

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

## Acknowledgements

This work was supported by an Australian Research Council (ARC) Discovery Project (DP110101894) and by the ARC Centre for Coherent X-Ray Science (CE0561787). H.X. was supported by an ARC Discovery Early Career Researcher Award (DE130101628). D.K. was supported by an ARC Future Fellowship (FT110100513). F.H. was supported by the NSF of China (Grant No. 11574205, 11322438 and 11421064). Simulations were performed on the π supercomputer at Shanghai Jiao Tong University.

## Author contributions

H.X. and I.V.L. conceived of and designed the experiment. H.X., X.W. and A.A.-T.-N. performed the experiment and analysed the experimental data. Z.L. and F.H. performed the numerical simulations. All authors contributed to discussions and preparation of the manuscript.

## Additional information

**Competing interests:** The authors declare no competing financial interests.

