## [Peer Review File · Nature Communications]

Reviewers' comments:

Reviewer #1 (Remarks to the Author):

Review of 114258

This work presents experimental data on the ionization of H₂ and subsequent dissociation of the resulting ion H₂⁺. The goal is to study in detail the dissociation process in a time-resolved manner with particular attention being paid to where the electron is during the dissociation. The authors find that after 15 fs (corresponding to 8 a.u.) the electron cannot be moved from one nucleus to the other. From this they conclude that the bond has been broken at this point.

The results are very interesting and the data appears to be of high quality. Hence, the paper has potential for publication in Nature Communications. However, there are a number of issues that need to be resolved, first.

1) The authors place this work in a very general and fundamental context: directly observing bond breaking. However, I feel that this is really not appropriate. The authors need to define exactly what they mean by bond breaking and justify why their definition is a good one. For a chemist not interested in strong field physics, I think bond breaking simply means that an internuclear coordinate goes to infinity. What the authors are measuring is how far apart can the nuclei get before a strong laser field of a particular pulse duration and wavelength can no longer drive the electron between the nuclei. This is interesting from a strong field control point of view, but not nearly as fundamental as they make it out to be and it is not the common sense meaning of bond breaking. The first laser pulse has broken the bond – the second pulse does not have the ability to reverse the bond breaking, it just selects which nucleus the electron ends up on. It also does not even mean that the electron is localized. It is only localized in the sense of this particular interaction cannot change the outcome.

2) The most important piece of data is shown in Fig. 2c. Around line 105 of the paper, the authors discuss this figure and the interference between the two laser pulses. I am concerned about the actual pulse duration and transform limit of these pulses. They differ in intensity by a factor of 10. However, even a difference in a factor of 100 can create a substantial modulation of the laser intensity: if the intensity ratio is 100:1, the electric field ratio is 10:1. This will produce a ratio of 11:9 in electric field for constructive and destructive interference. This translates into 121:81 in intensity, or about a 50% modulation. This is huge. In other words, if they have a pedestal on their pulse at the 0.1% level, they will get substantial modulation of the intensity. (Consider the pump pulse to have a peak intensity of 1000 and the probe 100. If the pump has a 20 fs pedestal with an intensity of 1, the peak probe to pump pedestal ratio is 100:1, leading to strong interference.) So, we need to see some evidence of how clean their pulses are. For example, just a measurement of the energy as a function of pulse-probe delay will reveal how far out in time delay that interference is significant.

3) The H₂⁺ system starts coherently. There is some superposition of states that ultimately places the electron on one side of the molecule. Somewhere, there is an implication that this coherence is lost, otherwise, there is nothing special going on – the system would just evolve according to the TDSE. I wonder when and how the coherence is lost. Either there is a very rapid accumulation of phase which is beyond the detection sensitivity of the experiment or there is a dissipative process, like ionization. Have the authors thought about this? Finally, the authors do not provide a convincing mechanism for changing the asymmetry with the probe pulse after 5 a.u., as there is no longer the possibility for a resonant interaction with their laser pulse. The Frustrated tunneling model is not convincing, as it likely has a low cross section and cannot explain the bulk of the data. I think the authors need to take a look at Phys. Rev. Lett. 110, 073002 (2012) where it was demonstrated with 1w2w laser pulses, that asymmetry in dissociation can be induced through asymmetric *depletion* in the enhanced ionization region.

In summary, this work raises important questions and provides interesting data. However, the depth of analysis is not sufficient to warrant publication in Nature Communications, as it currently stands.

Reviewer #3 (Remarks to the Author):

The work by Xu et al. is an important experimental contribution in which, by using CEP-stable few-cycle pump-probe IR laser pulses and coincidence measurements for the bond breaking process in the H_2^+ molecule, they are able to monitor and control electron localization in the dissociating molecule. In particular, the authors show that electron localization happens 15 fs after the initial preparation of the H_2^+ molecule by ionization of neutral H_2 , when the internuclear separation reaches 8 a.u., and they consider this is the moment in which the chemical bond is broken and the molecule breaks apart into a proton and a hydrogen atom. They use state-of-the-art CEP-stable few-cycle pump-probe IR laser pulses (about 5 fs) in combination with 3D imaging in coincidence using a reaction microscope. In general, the results and discussions are convincing. However, there are speculations about some of the results occurring at longer delay times over 15 fs. More importantly, the work lacks of any theoretical simulation of the bond-breaking and electron localization process for a molecule such as H_2^+ , for which accurate theoretical simulations are nowadays feasible. I would recommend publication of this work in Nature Comm. if theoretical simulations of the important processes observed experimentally could be included.

The authors claim in page 4, last paragraph, that they observe enhanced ionization with a correlated double ionization channel with KER of 5-6 eV. However, apparently these data have not been shown in the paper. Can the authors show these results?

Some references should be included when referring to REMI in page 3 and in the methods section and also some references should be included in the methods section for other technical details.

In page 3, last paragraph, ATD and 3PD should be defined for the general reader. The same in pages 4 and 5 for EUV.

In summary, I consider this paper could be published once theoretical simulations are included to confirm the experimental observations for a system such as H_2^+ , which is feasible from the theoretical point of view.

Reply to Reviewer 1:

This work presents experimental data on the ionization of H₂ and subsequent dissociation of the resulting ion H₂⁺. The goal is to study in detail the dissociation process in a time-resolved manner with particular attention being paid to where the electron is during the dissociation. The authors find that after 15 fs (corresponding to 8 a.u.) the electron cannot be moved from one nucleus to the other. From this they conclude that the bond has been broken at this point.

The results are very interesting and the data appears to be of high quality. Hence, the paper has potential for publication in Nature Communications. However, there are a number of issues that need to be resolved, first.

We thank the referee for this nice summary of our manuscript. We also appreciate the referee's positive and useful suggestions. We answer all questions one by one as following.

1) The authors place this work in a very general and fundamental context: directly observing bond breaking. However, I feel that this is really not appropriate. The authors need to define exactly what they mean by bond breaking and justify why their definition is a good one. For a chemist not interested in strong field physics, I think bond breaking simply means that an internuclear coordinate goes to infinity. What the authors are measuring is how far apart can the nuclei get before a strong laser field of a particular pulse duration and wavelength can no longer drive the electron between the nuclei. This is interesting from a strong field control point of view, but not nearly as fundamental as they make it out to be and it is not the common sense meaning of bond breaking. The first laser pulse has broken the bond – the second pulse does not have the ability to reverse the bond breaking, it just selects which nucleus the electron ends up on. It also does not even mean that the electron is localized. It is only localized in the sense of this particular interaction cannot change the outcome.

We thank the referee for this comment. It is true that there is no single universally accepted definition of a chemical bond and therefore no consensus view on what constitutes bond breaking. It is also true that some chemists may adopt the extreme view that a bond is broken only when an internuclear distance goes to infinity. We consider such definition not very useful for understanding detailed dynamics of chemical processes. In fact, the whole point of our manuscript is to learn more about such detailed dynamics. We believe that defining a chemical bond on the basis of electron sharing between the two atoms is as good a definition as any since a loss of electron sharing due to electron localization on one atom is the main signature of bond breaking. We understand that this could be controversial and not universally accepted and to avoid controversy we have removed the “bond breaking” from the title, which now reads “Observing electron localization in a dissociating molecule in real time”.

The referee suggests that we observe not a fundamental phenomenon but an effect specific to particular pulse parameters. To answer that experimentally we would have to repeat our experiment for a series of different pump and probe pulses, which would be an immense undertaking. Instead, we performed a numerical simulation, which very well reproduced our experimental observations with the same pulse parameters, and then repeated those calculations for a number of different pulse parameters, with intensity of the probe pulse, for example, varying by almost an order of magnitude. Qualitatively the behavior remains the same, as does the 15 fs electron localization time, as can be seen from the figure r1 below.

Figure_r1. The top row is the calculated energy resolved asymmetry as a function of delay for a fixed probe intensity of $6e13$ W/cm² and various pump intensity of (a) $6e14$ W/cm², (b) $4e14$ W/cm², (c) $3e14$ W/cm² and (d) $2e14$ W/cm². The bottom row is for a fix pump intensity of $6e14$ W/cm² and various probe intensity of (e) $2e14$ W/cm², (f) $1e14$ W/cm², (g) $6e13$ W/cm² and (h) $3e13$ W/cm².

We revised our paper by adding the description of our numerical simulation, the comparison of measurements and calculations (see revised figure 2) and an appropriate discussion. We believe that robustness of our results in respect to varying pulse parameters supports our claim that we observe a signature of a fundamental molecular dynamics rather than a pulse specific effect.

2) *The most important piece of data is shown in Fig. 2c. Around line 105 of the paper, the authors discuss this figure and the interference between the two laser pulses. I am concerned about the actual pulse duration and transform limit of these pulses. They differ in intensity by a factor of 10. However, even a difference in a factor of 100 can create a substantial modulation of the laser intensity: if the intensity ratio is 100:1, the electric field ratio is 10:1. This will produce a ratio of 11:9 in electric field for constructive and destructive interference. This translates into 121:81 in intensity, or about a 50% modulation. This is huge. In other words, if they have a pedestal on their pulse at the 0.1% level, they will get substantial modulation of the intensity. (Consider the pump pulse to have a peak intensity of 1000 and the probe 100. If the pump has a 20 fs pedestal with an intensity of 1, the peak probe to pump pedestal ratio is 100:1, leading to strong interference.) So, we need to see some evidence of how clean their pulses are. For example, just a measurement of the energy as a function of*

pulse-probe delay will reveal how far out in time delay that interference is significant.

We thank the referee for this comment. That is true that interference of pump and probe pulses remains significant even for large delays and it can be seen in continued modulations of asymmetry up to 40 fs delays. The interference between pedestal of pump and probe pulse has been investigated in our previous work (ref. 16) on enhanced ionization of H_2^+ . As shown in ref. 16, such interference can cause periodic modulation (period equals to laser cycle) of the enhanced ionization yield. By comparing the simulated and measured delay dependence, the pedestal intensity is estimated to be $3e12$ W/cm² when peak intensity is $6e14$ W/cm² (see ref. 16 methods). That interference does not affect our main result and conclusions. We included the pedestal in our simulations and found that the 15 fs cut-off time for electron localization (our main result) was completely insensitive to it.

Nevertheless, we were very careful to use clean transform-limited pulses. Outside COLTRIMS, both the pump pulse and probe pulse are individually diagnosed by a commercial Michelson interferometer based auto-correlator (Femtometer, few-cycle pulse autocorrelator), where both pulses have a measured transform limited pulse duration of 5–6 fs. To make sure the pulses duration is still minimal inside COLTRIMS, we carefully adjusted a pair of fused silica wedges to control the chirp of the few-cycle pulse so that the KER of sequential double ionization of H_2 is maximized (~ 10 eV), while the enhanced ionization yield of H_2^+ (with KER ~ 5 -6 eV) is minimized. As the Mach-Zehnder interferometer, which is used to produce pump-probe pulse pair, has a balanced GDD in its two arms, pump and probe pulses can simultaneously reach their transform limited duration in COLTRIMS.

3) The H_2^+ system starts coherently. There is some superposition of states that ultimately places the electron on one side of the molecule. Somewhere, there is an implication that this coherence is lost, otherwise, there is nothing special going on – the system would just evolve according to the TDSE. I wonder when and how the coherence is lost. Either there is a very rapid accumulation of phase which is beyond the detection sensitivity of the experiment or there is a dissipative process, like ionization. Have the authors thought about this?

We don't think the coherence of H_2^+ system is ever truly lost. During the dissociation, the electron is in the superposition of $1s\sigma_g$ and $2p\sigma_u$ states. After the probe pulse is finished, the electron wave packets can be written as $\Psi(t) = c_1|1s\sigma_g\rangle e^{i\delta - iE_1t} + c_2|2p\sigma_u\rangle e^{-iE_2t}$, where δ denotes a certain phase difference and $E_{1,2}$ is the eigen energy of the two states. By projecting $\Psi(t)$ onto $\frac{1}{\sqrt{2}}(|1s\sigma_g\rangle \pm |2p\sigma_u\rangle)$, we obtained the amplitude for the electron on each nucleus (denotes as the left and right side), i.e., $|\psi_{left}\rangle = \frac{1}{\sqrt{2}}(c_1 e^{i\delta - iE_1t} + c_2 e^{-iE_2t})$, $|\psi_{right}\rangle = \frac{1}{\sqrt{2}}(c_1 e^{i\delta - iE_1t} - c_2 e^{-iE_2t})$. When

the internuclear distance becomes large, E_1 and E_2 become constant and degenerate. Because of the energy degeneracy, $\langle \psi_{left} | \psi_{left} \rangle$ and $\langle \psi_{right} | \psi_{right} \rangle$ will not evolve with time any longer. Here by degeneracy we mean that Born-Oppenheimer energies of the two states of opposite parity are separated by less than the non-BO coupling terms of the full Hamiltonian.

*Finally, the authors do not provide a convincing mechanism for changing the asymmetry with the probe pulse after 5 a.u., as there is no longer the possibility for a resonant interaction with their laser pulse. The Frustrated tunneling model is not convincing, as it likely has a low cross section and cannot explain the bulk of the data. I think the authors need to take a look at Phys. Rev. Lett. 110, 073002 (2012) where it was demonstrated with 1w2w laser pulses, that asymmetry in dissociation can be induced through asymmetric *depletion* in the enhanced ionization region.*

We thank the referee for pointing this out and for bringing that very interesting and relevant paper to our attention. We completely accept the referee's view, that frustrated tunneling mechanism cannot account for the asymmetry modulation after 5 a.u.. After analyzing our data, we find that the modulation is most likely caused by the proton signal from the enhanced ionization channel. When the delay is around 30-40 fs, the KER of EI is around 3 eV, which is similar to that of the dissociative channel. Part of the ionized fragments was inevitably counted with the dissociative events in calculating the asymmetry because the filter we used cannot completely exclude the double ionization events (see methods for details). The proton KER spectra with dissociation filter (fig_r2 (a)) and with double ionization filter (fig_r2 (b)) demonstrate that effect between two channels as discussed above. It is also quite possible that EI is itself asymmetric in a manner described in that reference for I_2 . In that case the asymmetry would be affected by EI even if we could cleanly separate the dissociation channel. These effects are not included in our model calculations of asymmetry. We completely revised our explanation in the article and cited the mentioned work as reference [17] in the revised manuscript.

Figure_r2. The delay dependent KER spectrum when applying (a) filter for selecting dissociation

channel, and (b) filter for selecting double ionization channel. The white dashed line in two subfigures shows the decreasing KER of enhanced ionization channel with increasing delay.

In summary, this work raises important questions and provides interesting data. However, the depth of analysis is not sufficient to warrant publication in Nature Communications, as it currently stands.

We hope that we could convince the referee that with inclusion of a theoretical simulation and other revisions the depth of analysis is improved and our revised paper could be published in Nature Communications.

Reply to Reviewer 3:

The work by Xu et al. is an important experimental contribution in which, by using CEP-stable few-cycle pump-probe IR laser pulses and coincidence measurements for the bond breaking process in the H₂⁺ molecule, they are able to monitor and control electron localization in the dissociating molecule. In particular, the authors show that electron localization happens 15 fs after the initial preparation of the H₂⁺ molecule by ionization of neutral H₂, when the internuclear separation reaches 8 a.u., and they consider this is the moment in which the chemical bond is broken and the molecule breaks apart into a proton and a hydrogen atom. They use state-of-the-art CEP-stable few-cycle pump-probe IR laser pulses (about 5 fs) in combination with 3D imaging in coincidence using a reaction microscope. In general, the results and discussions are convincing. However, there are speculations about some of the results occurring at longer delay times over 15 fs. More importantly, the work lacks of any theoretical simulation of the bond-breaking and electron localization process for a molecule such as H₂⁺, for which accurate theoretical simulations are nowadays feasible. I would recommend publication of this work in Nature Comm. if theoretical simulations of the important processes observed experimentally could be included.

We thank the referee for his/her very positive comments of our manuscript. We answer all his/her questions below.

The authors claim in page 4, last paragraph, that they observe enhanced ionization with a correlated double ionization channel with KER of 5-6 eV. However, apparently these data have not been shown in the paper. Can the authors show these results?

We addressed this issue in our answer to Reviewer 1's last comment and present the time-dependent KER spectra after filtering for single and double ionization in figure r2 of this response letter. We added a discussion explaining it to the Experimental Scheme subsection of the Methods. We feel that adding a whole new figure to the paper just to emphasize that point is not warranted.

Some references should be included when referring to REMI in page 3 and in the

methods section and also some references should be included in the methods section for other technical details.

We thank referee for this excellent suggestion. In the revised version, we added two reviews which introduce the principle of REMI: ref 18 and ref 19 in the revised manuscript.

We also added two references for introducing intensity calibration method used in our experiment (ref 20 and ref 21 in the revised manuscript).

In page 3, last paragraph, ATD and 3PD should be defined for the general reader. The same in pages 4 and 5 for EUV.

We thank the referee for pointing out this problem. We defined explicitly the ATD, 3PD and EUV in revised manuscript.

In summary, I consider this paper could be published once theoretical simulations are included to confirm the experimental observations for a system such as H_2^+ , which is feasible from the theoretical point of view.

We took this suggestion seriously as we ourselves felt that a theoretical simulation was one missing piece to make this study complete. As suggested by the referee, in the revised version, we included the theoretical simulations. The simulation results shown in Fig. 2 (d) and (e) support the main observations in the experiment. The simulation details are described in the Methods section. Another important result of the simulation is that our main discovery of 15 fs cut-off time for electron localization is robust in respect to laser intensity variations over almost an order of magnitude. That supports our claim of observing a fundamental molecular dynamics rather than some pulse parameter dependent effect.

Besides the numerical simulation shown in the revised manuscript, we also used the two-state model to simulate the dissociation process. In that simulation, only the $1s\sigma_g$ and $2p\sigma_u$ are included. Very similar numerical conclusions were reached.

The remaining mismatch between the simulation and experiment mainly comes from the inaccurate description of the initial nuclear wave packet of H_2^+ . In the simulation, we used the Franck-Condon approximation to describe the nuclear wave packet just before the pump pulse. However, during the single ionization of H_2 in the experiment, the freed electron and the produced H_2^+ are entangled, thus the interference of the ionized electron wave packet emitted at different instants will map in the produced H_2^+ wave packet. In other words, the nuclear wave packets produced at different times by the pump pulse will overlap together, and one should not simply sum all the nuclear wave packets either fully coherently or full incoherently without considering

the freed electron. To precisely simulate the experiment, one should simulate H_2 by including two-electron dynamics. Nevertheless, our simplified model has already been able to explore the most important experimental feature.

REVIEWERS' COMMENTS:

Reviewer #1 (Remarks to the Author):

I think the authors adequately addressed the concerns raised in the first review. However, I found one thing missing: in the simulations, they used a soft-Coulomb potential using a softening potential $\alpha(R)$. But, they did not provide the function or even its general form. Simulation results and energy levels can be quite sensitive to this. I don't expect that there is a problem, here, but if someone wanted to reproduce their results or build on them, they could not. I think it is important to add this information. Otherwise, I recommend publication.

Reviewer #3 (Remarks to the Author):

After a careful evaluation of the revisions included in the revised version of the manuscript by Xu et al., I recommend publication of this revised version in Nature Comm. The authors answered satisfactorily to all questions raised by the two reviewers, with appropriate justifications and proper additions and corrections. The authors also included the requested numerical simulations and reshape the discussions accordingly. This way the manuscript has been very much improved and thus it is now suitable for publication in Nature Comm.

We would like to thank the Reviewers for their invaluable help with improving our manuscript. In his/her last communication Reviewer #1 made a suggestion about the soft-core function used in our numerical model. Specifically he wrote:

“I think the authors adequately addressed the concerns raised in the first review. However, I found one thing missing: in the simulations, they used a soft-Coulomb potential using a softening potential $\alpha(R)$. But, they did not provide the function or even its general form. Simulation results and energy levels can be quite sensitive to this. I don't expect that there is a problem, here, but if someone wanted to reproduce their results or build on them, they could not. I think it is important to add this information. Otherwise, I recommend publication”.

We agree that this information could be useful for some readers. We added the following text to the manuscript (in red):

“... where $\alpha(R)$ is the R -dependent soft-core-function, which is determined by fitting the calculated energies of the ground and the first excited state of H_2^+ to their real Coulomb values. That function can be closely approximated by $\alpha(R) = 0.9 + 0.5806 \times \exp(-0.5876 R)$ and it monotonously decreases from 1.5 to 0.9 as R increases”.